# Canalized light creates directional and switchable surface structures in vanadium dioxide

Daniel Kazenwadel [1,2], Noel Neathery[1,2] & Peter Baum [1] ✉

Materials with switchable nanostructured surfaces enable optical and electronic functionalities beyond those of natural materials. Here we report the creation of self-organized, re-writable, laser-induced surface structures in single-crystalline vanadium dioxide. We discover anisotropic features caused by canalized surface plasmon polaritons that can only propagate along one crystal axis. The nanostructures remain mostly single-crystalline and preserve the material's sharp metal-to-insulator transition, enabling femtosecond switching by temperature or light.

The controlled manipulation of light is essential for optical technologies and the backbone of modern communication technology. In particular, switchable materials that can change their optical properties on femtosecond time scales are useful for ultrafast optics and information processing. To unleash maximum functionality, it is important to combine complex optical materials with tailored nanostructures at sub-wavelength dimensions to produce metamaterial effects beyond the capabilities of the original material. For example, metamaterials are useful for consumer electronics[1,2], efficient light harvesting with solar cells[3], or electronic circuitry[4].

When irradiating a solid material with intense linearly polarized laser light, there is the possibility of inducing surface reliefs termed laser-induced periodic surface structures (LIPSS)[5–24]. These structures typically have periodicities slightly below the wavelength of the excitation laser and are aligned either parallel or orthogonal to the polarization of the incident laser light[5]. Such surfaces have various applications in friction reduction[5], nanophotonics[6,7], photocatalysis[8], dewetting[9,10] or medicine[11]. However, laser-modified materials usually become amorphous[12,13] or polycrystalline[14], and they are rarely re-writable or switchable, although such features would be highly beneficial for active meta-optics, nanophotonic computation, or ultrafast control of light.

In this work, we investigate surface structures in vanadium dioxide, a strongly correlated material that is famous for its enigmatic and complex insulator-to-metal transition[25–27] and structural dynamics[28–30] which enable, for example, ultrafast photoelectric switches[31,32], thermochromic windows[33–35], ultrasensitive bolometers[36], neuromorphic computing[37,38] or metamaterials[39–41]. Its refractive index is highly anisotropic along the different crystal directions[41,42]. After laser treatment, we discover on our sample different kinds of periodic surface structure including one caused by canalized light. All nanostructures remain single crystalline and can be switched on ultrafast timescales.

## Results

For our experiments, we grow bulk single crystals with a surface roughness below 10 nm via thermal decomposition of $V_2O_5$ at 975 °C under an argon atmosphere[43]. Irradiation is provided by a femtosecond laser at a center wavelength of $\lambda = 1030$ nm, a pulse duration of $\tau = 300$ fs, and a repetition rate of 0.1–200 kHz (see methods).

Figure 1a shows scanning electron microscopy images of nanostructures at a fluence of 400 mJ/cm². We see periodic surface structures aligned orthogonal (±15°) to the polarization of the laser light (red arrow). Figure 1b shows a side view of the grooves and reveals a nearly sinusoidal shape with a periodicity of 880 nm and a depth of 390 nm. When we rotate the incoming laser polarization (Fig. 1c), the grooves maintain this periodicity and quality but change their direction.

All fabricated nanostructures remain single-crystalline (except some occasional twinning, see below). Supplementary Fig. 1 shows electron back-scattering diffraction images from the fabricated area. We use an electron beam close to grazing incidence (70°) and align it orthogonal to the grooves. We therefore probe only the ridges, not the substrate. The presence of all expected Kikuchi diffraction lines shows that the nanostructured material is still mostly single-crystalline. We conclude that the grooves in Fig. 1 are probably ablated under preservation of the single-crystal nature of the humps.

---

[1]Universität Konstanz, Fachbereich Physik, 78464 Konstanz, Germany. [2]These authors contributed equally: Daniel Kazenwadel, Noel Neathery.
✉e-mail: peter.baum@uni-konstanz.de

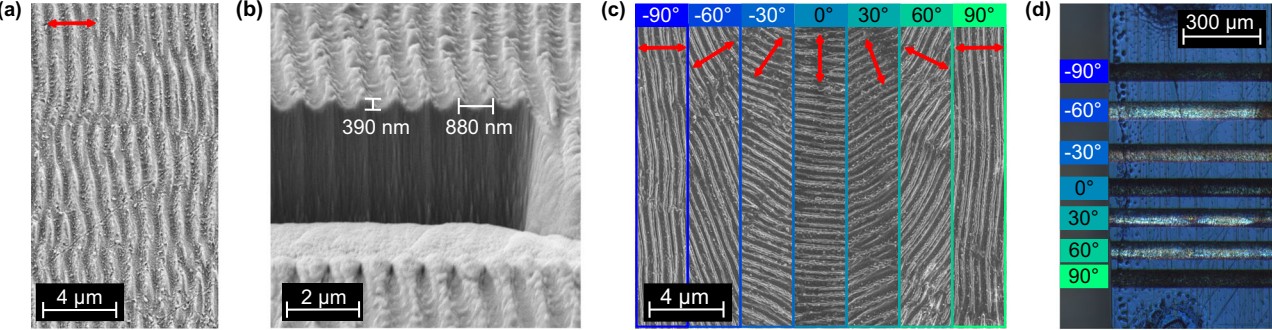

**Fig. 1 | Line-scans on a VO₂ single crystal. a** SEM images of the structures written with a fluence of 400 mJ/cm². **b** FIB-cut through one of the structures. The grooves have a depth of 390 nm and a periodicity of 880 nm, slightly below the laser's wavelength of 1030 nm. **c** Polarization dependency; when the laser polarization is rotated (red arrows), the grooves align always orthogonal to it. **d** Cross-polarized light microscopy image of a macroscopic bulk crystal with differently oriented LIPSS. Different grating orientations change the incoming light's polarization, allowing it to pass through the analyzer. The wavelength dependence of refraction can be seen as different colors of the individual lines. Intensity scale, see methods.

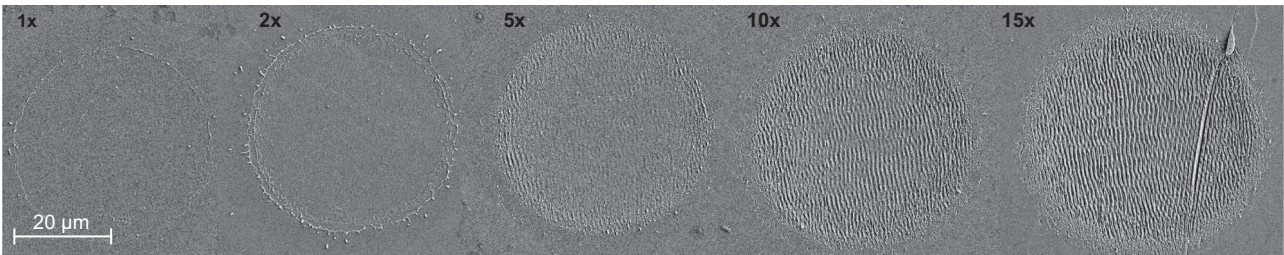

**Fig. 2 | Shot-to-shot evolution of LIPSS.** Left to right. SEM images of LIPSS formation after increasing numbers of incoming laser pulses (from left to right) with a fluence of 150 mJ/cm². We see how the first pulse creates a single ring of roughened structure. A periodic pattern nucleates on the outside and then propagates to the inside with each successive pulse until after 15 shots a well-formed periodic structure over the full area of the laser spot can be seen. Intensity scale, see methods.

Figure 1c shows that the laser-induced surface grooves are always orientated in perpendicular direction to the laser polarization. Zero degrees denotes here a polarization parallel to the $a_m/c_r$ axis of the high-temperature rutile phase of VO₂. This axis can easily be identified in the experiments because the crystals are long needles along $a_m/c_r$[43]. Figure 1d shows a cross-polarized optical microscopy image of the surface in millimeter dimensions. The polarizer is aligned parallel to the rutile c-axis of the crystal, and the analyzer is orthogonal to it. We see different colorings and intensities of the reflected light. The reflected intensity is lowest when the grooves are aligned parallel to either the polarizer or the analyzer (0° and 90°). At other angles, the grooves slowly turn the polarization in a stepwise way, resulting in higher measured reflected intensities. Outside of the grooves, blue light is observed because VO₂ in its low-temperature, insulating monoclinic phase is naturally birefringent along the monoclinic $c_m$ axis that is aligned at an angle of 122.6° with respect to the long axis of the rutile metallic material[42,43].

Figure 2 shows scanning electron microscopy images of the shot-to-shot formation of our grooves. We see that the first single laser pulse only roughens up the surface at the edge of the spot, probably at the highest intensity gradient. The second laser pulse then starts the nucleation of periodic surface structures inwards from this edge. With an increasing number of laser shots, the pattern propagates to the center of the spot. After ~15 shots, the whole area is covered by regular and stable nanometer grooves (compare Fig. 1). Cooling after each laser excitation requires only roughly 100 ns[44] and the structure can therefore be written within sub-microsecond times.

Laser-induced periodic surface structures are typically explained by surface plasmon polariton waves which form a standing wave[7,19,20] which is, in our experiment, phase-stabilized on the circular initial

defect ring (see Fig. 2). Surface plasmon polaritons have forward and backward electric fields and are therefore emitted from dipolar defects predominantly along the incident polarization direction. The resulting standing waves from large or multiple defects or interferences between the plasmon and the incident light cause a periodic field enhancement that partially melts or ablates the material[16]. Further laser shots enhance these dynamics and propagate the periodic structure all the way through. Other theories invoke more complex electromagnetic surface waves[22] or self-organization of a softened material[23,24].

However, in its low-temperature insulating state, VO₂ does not support surface-plasmon polaritons because it is an insulator and its dielectric constant is not negative[41,42]. We argue that we almost fully convert our VO₂ surface into its metallic phase during each laser shot because our laser fluences of > 100 mJ/cm² exceed the latent heat of 3.1 mJ/cm²[44–47] and our pulse duration of 300 fs is about three times longer than the 80-fs switching time of the metal-to-insulator transition[48]. Therefore, the material partially transforms into a metal before the energy of the remaining laser pulse triggers a now-allowed surface plasmon polariton wave. Also, we excite VO₂ with photons of 1.2 eV which is above the bandgap of 0.6 eV[49,50]. The resulting non-equilibrium carrier-hole pairs live for ~200 fs[48] and also contribute to a transient metallicity of the refractive index, facilitating surface plasmon polariton formation.

Our surface structures can repeatedly be overwritten in a deterministic way. In the experiment (Fig. 3), we first write one horizontal line of grooves at a polarization of 116° (arrow 1). Then we write at another polarization of 64° a vertical line of grooves that crosses the original one. We see in the scanning electron microscopy images that the vertical line overwrites the original structure (Fig. 3a). When we re-

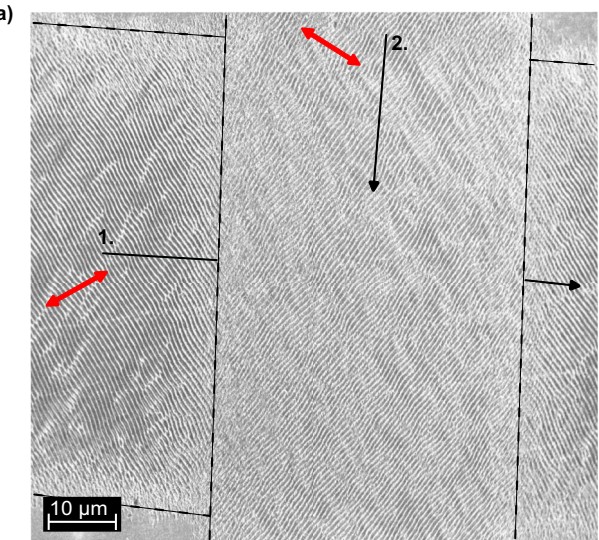

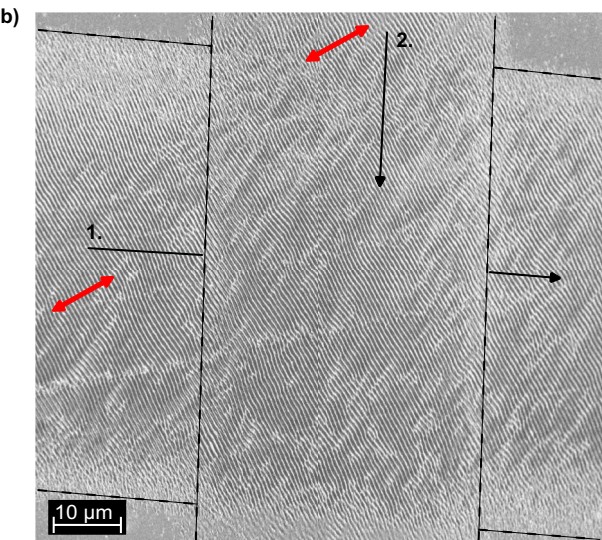

**Fig. 3 | Rewriting the LIPPS.** SEM-images of two line-crossings. A first laser (1.) writes with 116° polarization, indicated with the red arrow, and then a second laser (2.) writes at either 64° (**a**) or 116° (**b**). We see, that the vertical lines overwrite the previously written horizontal lines in case of different polarization. The dashed lines mark the edge of the written area. Intensity scale, see methods.

write with the same polarization as initially, the original orientation is maintained, and the newly written grooves align in-phase to the original ones (Fig. 3b).

We also can control the ripple distance; Fig. 4 shows the results. From left to right, we decrease the laser fluence from 140 mJ/cm² over 120 mJ/cm² down to 90 mJ/cm². The upper panels show scanning electron microscopy images of the resulting surface structures, and the lower panels show a zoom. Figure 4a shows the already reported low-spatial-frequency pattern with a periodicity of ~900 nm. In contrast, Fig. 4b shows at the outer rim a periodicity of ~200 nm. This result has an opposite fluence dependency than in many previous reports which concluded that the high-spatial-frequency ripples seen at the outer rim might be caused by optical harmonic generation[51,52]. In our single-crystalline vanadium dioxide this is clearly not the case, because the high-spatial-frequency ripples occur at lower fluences than the low-spatial-frequency grooves. We guess that the combination of lower photo-doping and potentially a lower amount of metallic VO₂ opens up another self-stabilizing channel for surface plasmon polariton formation at a much shorter wavelength. Steel has similar

features of unknown origin[53]. At about half of the focus diameter, we see a change from high-spatial-frequency to low-spatial-frequency grooves. This transition between the two possible patterns is almost abrupt because only the optical mode with the highest self-organization gain will survive and dominate the macroscopic growth. All these patterns are aligned orthogonal to the laser polarization.

However, when the writing fluence is lowest (Fig. 4c), we see besides the conventional groove pattern (dashed line) a new sub-harmonic structure (solid line). The wavelength is ~2 μm which is approximately two times the laser wavelength. To our surprise, the alignment of this subharmonic structure is independent of the polarization of the generating laser light. Figure 5 shows scanning electron microscopy images as a function of laser polarization (red arrows). We see that the conventional surface structures (dashed lines) rotate with polarization but the subharmonic structure (solid lines) maintains a constant direction of −43 ± 3° for all applied laser fields. The lower panels show two-dimensional Fourier transformations of the real-space results. The conventional structures appear as spots that rotate with laser polarization (dashed arrows). The tails indicate a non-sinusoidal shape of the grooves (compare Fig. 1). In contrast, the sub-harmonic structure appears as spots at closer inverse distances (solid arrows) and their orientation always stays the same. The direction of the subharmonic features is even better defined (±3°) than the direction of the conventional high-frequency grooves (±11°).

To investigate the nature of this phenomenon, we record a scanning electron microscopy image (Fig. 6a) and an electron back-scattering diffraction pattern (Fig. 6b) of the sub-harmonic grooves under preservation of absolute sample orientation. We can therefore relate a measured sub-harmonic groove orientation with the low-temperature, monoclinic crystal structure.

We grow our VO₂ crystals at high temperatures and the macroscopic surface for laser machining is a (110) surface of rutile VO₂. Cooling down into the monoclinic phase causes a doubling of the unit cell. Pairs of vanadium atoms slightly tilt in the rutile (010) or (100) planes and form dimers. Consequently, the $c_m$ axis of the low-temperature monoclinic phase lies in one of these two high-temperature planes[48]. This dimerization of the vanadium atoms breaks symmetry in four potential directions, leading to new crystal axes $c_m \approx -c_r \pm a_r$ or $c_m \approx -c_r \pm b_r$[42,48,54] where the subscripts $m$ and $r$ refer to the monoclinic and rutile structures, respectively. The drawings in Fig. 6a show the two possible projections of the monoclinic $a_m$, $b_m$, and $c_m$ axes onto the surface of our macroscopic crystal. We see that $c_m$ obtains an angle of ±40.6° with respect to the $c_r$ and $a_m$ axis. Back-scattering electron diffraction (Fig. 6b) confirms that the lattice and atoms in our crystal surface are indeed aligned that way.

Our single-crystal surface of monoclinic VO₂ therefore can have differently oriented twin domains. With our cross-polarized optical microscopy, we now search an area with two domains and write laser grooves (inset of Fig. 6a); the dashed blue lines mark the measured domain boundaries. The main part of Fig. 6a shows a scanning electron microscopy image of the laser-written sub-harmonic grooves. In the upper part, the sub-harmonic grooves align at a measured angle of 43 ± 3° and in the lower part at −43 ± 3°, indicated by dashed white lines. These results show that laser-induced surface structures can indeed grow independently of polarization along distinctive directions on a crystal surface.

## Discussion

Monoclinic VO₂ is birefringent[42] and the anisotropy of the dielectric constant aligns with the $c_m$ axis, confirmed by optical anisotropy measurements (see Supplementary Fig. 2). Low-temperature monoclinic VO₂ is not a metal and therefore does not support surface plasmon polaritons in any direction. On the other hand, metallic high-temperature VO₂ would support plasmons polaritons but its

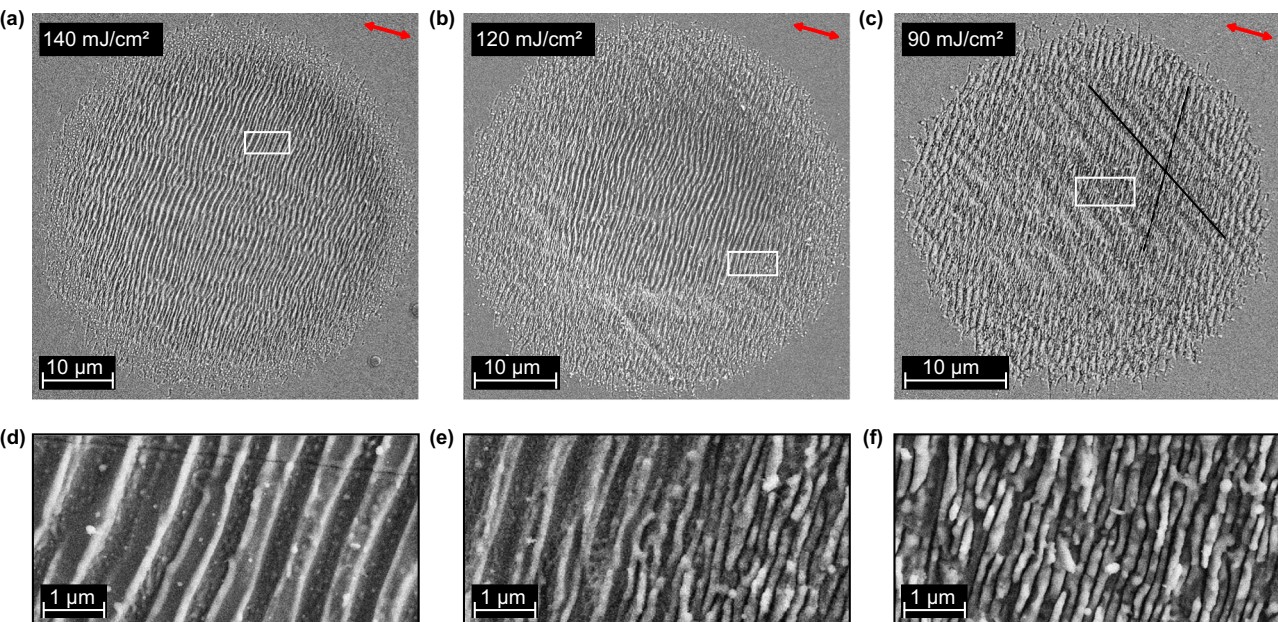

**Fig. 4 | Fluence dependence.** Scanning electron microscopy images with different fluences. **a** For a fluence of 140 mJ/cm²; we see 900-nm grooves oriented orthogonal to the polarization of the laser light. **b** For 120 mJ/cm², we see additional grooves with a periodicity of ~200 nm at the edge of the spot, where the fluence is lower than in the middle. **c** At 90 mJ/cm², a new periodic pattern, aligned at ~45°, appears. **d**–**f** Magnified views of the upper images; the white boxes mark the magnified areas. Intensity scale, see methods.

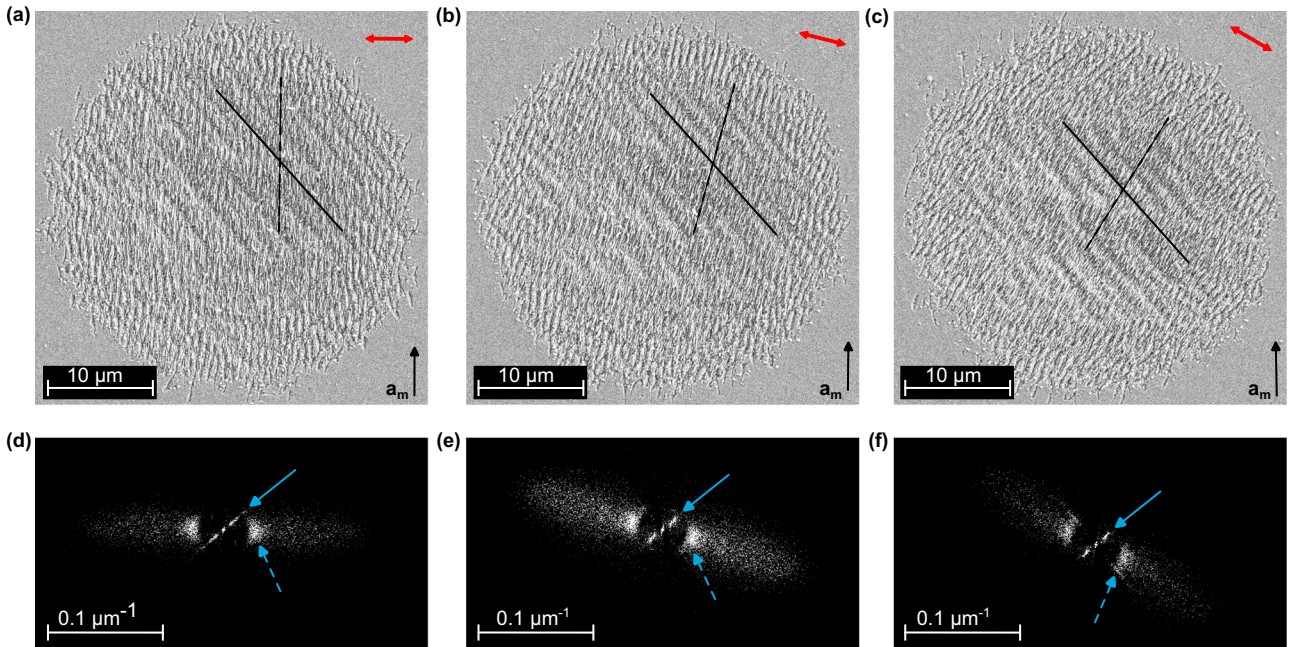

**Fig. 5 | Lattice-coupling of the subharmonic structure. a**–**c** Subharmonic structure (solid line) written with different laser polarizations (left to right). We see that the high-spatial-frequency LIPSS (dashed line) rotate with the polarization of the incoming laser (red arrows), the subharmonic structure however always stays at the same orientation of $-43 \pm 3°$ to $\mathbf{c_r}$ and $\mathbf{a_m}$ (black arrow). **d**–**f** Contrast-optimized Fourier transformation of the upper SEM images further confirms this result. Blue arrows indicate the structure features in the images above, dashed for high-spatial-frequency LIPSS and solid for the subharmonic structure. Intensity scale, see methods.

anisotropy along $\mathbf{c_r}$ (vertical in Fig. 6) does not align with the observed groove direction.

We argue that the two anisotropic but positive dielectric constants $\epsilon_\perp > \epsilon_\parallel$ of low-temperature $VO_2$[42,55] in perpendicular or parallel direction to $\mathbf{c_m}$ are both shifted by photo-doping towards the negative regime, but only $\epsilon_\parallel$ becomes negative in the early parts of the femtosecond laser pulse. Therefore, we have a transient regime in which $\epsilon_\parallel < 0$ and $\epsilon_\perp > 0$, at least for some time, before the transition is complete and electrons and phonons thermalize (~300 fs). Such a temporary state with anisotropic metallicity along $\mathbf{c_m}$ allows a surface plasmon polariton parallel to the $\mathbf{c_m}$ axis but does not allow surface wave propagation in the orthogonal direction[56]. This is the regime of

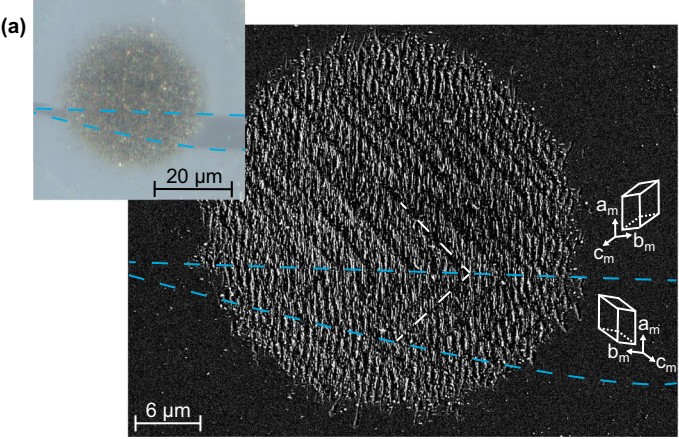
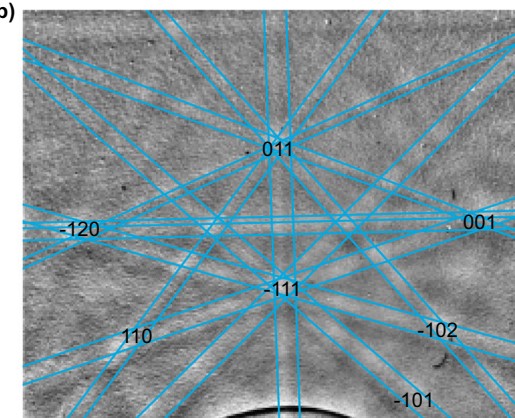

**Fig. 6 | Crystal orientation dependence of the subharmonic structure.**
**a** Subharmonic structures with different directions (dashed white lines). The microscope image taken with crossed polarizers (inset) confirms that the different patterns belong to different crystal orientations (separated by dashed blue lines).

**b** Electron back-scatter diffraction analysis proves the single-crystallinity of the written structure and further confirms this coupling. The most visible features are indexed with the corresponding Miller indices. Intensity scale, see methods.

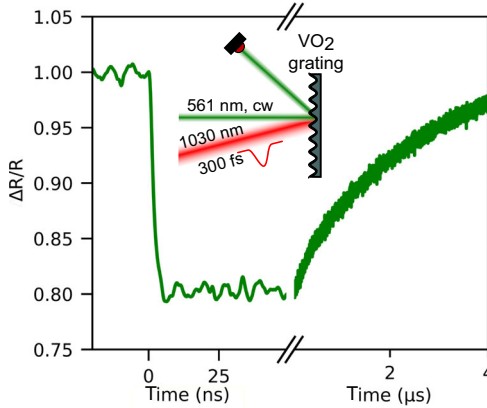

**Fig. 7 | Ultrafast laser-written diffraction grating.** We pump our regular periodic surface structure with an ultrafast laser and observe the time-dependent refracted intensity of a green continuous wave laser with a fast diode. We see that the grating switches with approx. 2 ns, limited by the speed of the diode.

canalized light[57–60], where a surface wave propagates without lateral dispersion for extended lengths. This explains the well-defined directionality of our sub-harmonic grooves (±3°). Only later in the femtosecond pulse, we push both dielectric constants into the negative regime and switch the material into the metal state. At this point, we can produce the conventional surface structures with a direction determined by the polarization. However, under special fluences, the canalized light can become strong enough to coexist with the conventional phenomenon (see Fig. 6a). Due to the transient nature of the canalized light in our experiments, it is unlikely that the directional grooves can be produced exclusively, but we expect that shorter femtosecond pulses and optimized wavelengths will be beneficial.

If our laser-machined $VO_2$ grooves still preserve all crystallographic relations to the original single crystal, they should remain switchable between insulator and metal with a sharp hysteresis curve by temperature or light. We can therefore expect that optical effects like diffraction or reflection can be turned on and off without the disadvantages observed in nanostructures with small grain size[61–63].

In the experiment (Fig. 7), we diffract green light from a continuous wave laser at a wavelength of 561 nm from a laser-written 900 nm grating and observe the diffracted intensity with a fast

photodiode. We then excite the grating with an ultrafast infrared pulse at a wavelength of 1030 nm to switch the vanadium dioxide from its insulating into the metallic phase. After laser excitation at $t = 0$ s, the diffracted intensity immediately drops by 20% with a bandwidth-limited speed. Because the optically induced insulator-to-metal transition of $VO_2$ occurs on ultrafast time scales of <100 fs[25,28–30,64], we expect the real switching to happen on a similar time scale. After about 4 μs, the nano-grating recovers to the original state. Therefore, our laser-generated surface structures can serve as re-writable and switchable materials for applications in nanophotonics.

These results show that laser-induced surface structures in single-crystalline and anisotropic media can be generated with a large variety of geometries by utilizing and optimizing the transient femtosecond properties of photo-doped materials. The tools of femtosecond laser science in combination with correlated solid-state materials can therefore be used to create laser-induced nanostructures at unprecedented complexity. In turn, measurements of fabricated structures can provide insight into the underlying optical dynamics of surface plasmon polaritons or canalized light, time-frozen into the solid material. For applications, single-crystalline laser-induced surface structures with adjustable periodicity along polarization-controlled or crystallographic directions may therefore become a useful tool for producing active metamaterials on macroscopic scales.

## Methods
### $VO_2$ crystals
We grow bulk single crystals via thermal decomposition of $V_2O_5$ at 975°C with liquid diffusion under an argon atmosphere. The surface roughness is below 10 nm, estimated from high-resolution scanning electron microscopy data[43].

### Optical anisotropy measurements
We use a 1030 nm laser with a focus size of ~10 μm to measure polarization-dependent reflectivity on $VO_2$ in the low-temperature phase for different crystallographic twins. The different low-temperature domains are identified using polarized light microscopy. The laser hits the sample at an angle of ~3°, close to normal incidence. Supplementary Fig. 2 shows the polarization-dependent reflectivity of a domain tilted to the right (dashed line) and a domain tilted to the left (solid line). The insets show the unit cell orientation of these twin domains. We see a maximum reflectivity in the direction of the two different $c_m$ axes. When the material is photodoped (arrows), the material becomes metallic (dotted line) in a faster and more

efficient way in a crystal direction along $c_m$. After the structural phase transition and electron-phonon relaxation (~300 fs), the anisotropy returns to that of the high-temperature phase[27].

## Irradiation parameters

We use a femtosecond laser (Pharos, Light Conversion) at a center wavelength of $\lambda = 1030$ nm, a pulse duration of $\tau = 300$ fs, and an adjustable repetition rate of 0.1–200 kHz. In Fig. 1, we use a fluence of 400 mJ/cm² and slowly line-scan the laser spot over the sample with a speed of ~2 mm/s at a repetition rate of 1 kHz. In Fig. 2, we use pulse trains of varying lengths with a fluence of 150 mJ/cm² at a repetition rate of 0.1 kHz. The number of shots increases from left to right. In Fig. 3, we use a fluence of 400 mJ/cm² at a repetition rate of 1 kHz and slowly (~2 mm/s) line-scan the laser spot over the sample. In Fig. 4, we use a repetition rate of 0.1 kHz. The fluences are 140 mJ/cm² in panel a, 120 mJ/cm² in panel b, and 90 mJ/cm² in panel c. The number of shots $N$ was adjusted so that a consistent and stable structure forms, resulting in $N = 20$ for panel a, $N = 75$ for panel b, and $N = 200$ in panel c. In Fig. 5 and Fig. 6, we use the same laser parameters as Fig. 4c. To rule out radiation remnants from the optical setup as a source of the sub-harmonic grooves, we rotated not only the polarization, but also the sample itself. In both cases the structures stay well-aligned with the crystal lattice. To produce the ultrafast diffraction grating depicted in Fig. 7, we write a regular grating with an area of several mm² by scanning the laser spot over one of our bulk single crystals using a rectangular pattern at the same parameters as used in Fig. 4a.

## Ultrafast diffraction measurements

The pump laser has a center wavelength of 1030 nm, a pulse duration of 300 fs (Pharos, Light Conversion) and a repetition rate of 5 Hz which allows the sample to completely relax to room temperature after each laser excitation. This pump laser spot has a full width at half maximum of 300 μm and the fluence is 20 mJ/cm², far below writing threshold. The probe laser is a green continuous-wave laser (DPL 561 nm, Cobolt) at a wavelength of 561 nm and illuminates the grating at orthogonal incidence. The spot size is much smaller than that of the excitation laser. Some of the probe beam intensity is refracted at an angle of ~48° and collected with another lens onto a fast photodiode (S5973-02, Hamamatsu) whose signal is analyzed with a GHz oscilloscope (Wavesurfer 44MXS-B, LeCroy).

## Scanning electron microscope

All scanning electron microscopy images are taken with a scanning electron microscope (Gemini 500, Zeiss) at an acceleration voltage of 5 keV. The wedge for seeing the grooves under non-normal incidence (Fig. 1b) is produced by focused ion beam milling (CrossBeam 1540XB, Zeiss) using gallium ions at a current of 50 pA. Black and white in images and Fourier transforms denote low and high effective currents of secondary electrons after brightness and contrast optimization, respectively.

## Electron backscatter diffraction

The electron backscatter diffraction data for the determination of the material's single crystallinity and identification of the individual crystal axes is recorded with an Oxford Instruments EBSD detector in a scanning electron microscope (Gemini 500, Zeiss). A scheme of the setup is depicted in Supplementary Fig. 1. The electron beam has an energy of 20 keV and hits the sample at an angle of 70°, close to grazing incidence. The direction of the grooves is aligned orthogonal to the plane of incidence, and our experiment probes the material of the ridges and not the solid substrate. The backscattered electrons are detected by scintillation and analyzed (AZtecCrystal, Oxford instruments) to calculate the crystal indexing depicted in Fig. 6b and Supplementary Fig. 1b. We see clear Kikuchi lines that belong to the monoclinic low-temperature phase of $VO_2$. The partial indexing shown

in Fig. 6b confirms the orientations depicted in Fig. 6a. While all visible features can be matched with the simulation, we do not show all labels for clarity. Indexing with calculated diffraction patterns from higher vanadium oxides like $V_2O_5$ performs far worse or does not match at all. The slight blurring of the data is probably caused by the rough surface[65], the crystals are not oxidized[66].

## Data availability

All data are available from the corresponding author upon request.

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

## Acknowledgements

The authors acknowledge financial support by Evangelisches Studienwerk e.V. and Deutsche Forschungsgemeinschaft via SFB 1432. We

thank Roman Hartmann, Matthias Hagner and Chantal Kesting for help with the electron backscattering diffraction, and Jakob Müller for assistance with the optical anisotropy measurements.

## Author contributions

D.K., N.N., and P.B. performed research and wrote the paper.

## Funding

## Competing interests

The authors declare no competing interests.
