## [Transparent Peer Review file · Nature Communications]

Canalized Light Creates Directional and Switchable Surface Structures in Vanadium Dioxide

Corresponding Author: Professor Peter Baum

Version 0:

Reviewer comments:

Reviewer #1

(Remarks to the Author)

The manuscript by Kazenwadel *et al.* describes intriguing observations of laser-induced periodic surface structures (LIPSS) on VO₂ single crystals. The authors examine the build-up of the LIPSS after a number of laser shots, the orientation of the generated grooves relative to the laser polarization, the connection between their spatial periodicity and the laser fluence used, and the groove structure's capability to change the polarization of incoming light. The authors further report the rewritability of the LIPSS in different directions and find a different subharmonic structure registered to a specific crystal axis. The authors claim that the LIPSS remain single-crystalline based on electron back-scatter diffraction (EBSD) and hypothesize the presence of an anisotropy in the dielectric function of the transient metallic phase that could facilitate surface plasmon polariton formation for the LIPSS. With the produced grating-like structure, the authors suggest potential applications such as active metamaterials on macroscopic scales.

In my opinion, this is an important discovery with both fundamental and technological significance. The report appeals to broad readership from materials and engineering research fields, including those highly interested in the use of the correlated material VO₂. The results are reasonably analyzed and explained with a proposed mechanism for the LIPSS formation. Overall, I believe it is suitable for consideration of publication. However, there are a few major issues that need to be resolved first.

(i) The authors mention the highest intensity observed at ~45° on page 3 of the text but Figure 1d shows only the observations made with a 30° step size not covering the claimed angle.

(ii) The authors' language of "periodic surface structures aligned orthogonal to the polarization of the laser light" on page 2 appears to be very specific. However, upon a closer look at Figure 2, the grooves' tilt angles vary noticeably from the left side to the right. While the preferred orientation is clear, the results are not strictly like the authors' claim. Figure 1c also shows a certain angular range in each panel. Therefore, I suggest the authors to adjust their description.

(iii) The EBSD patterns in Figures 6 and S1 are used to claim the single-crystallinity of the LIPSS. However, there are Kikuchi bands that are not indicated in the figures; are they also produced from the same crystal orientation or actually by maybe twin structures? Do the authors have more evidence for single crystallinity? Would it be just preferred orientations? Figure 1b appears to show features of grains/islands/bumps that may be hard to consider as fully single-crystalline, unless they were produced as a result of materials indeed being ablated (the authors describe this probability on top of page 3) and no melting and then recrystallization (the authors mention partial melting on page 4).

Here I provide another possibility that the grooves are produced with some contributions from materials fast melting and recrystallization, producing a certain thickness of polycrystalline surface layer. Given that EBSD information depth is tens of nanometers, this alternative scenario can still produce the observed pattern, with the polycrystalline layer giving diffuse

scattering onto the ordered pattern from the deeper part of the groove structure. How would the authors rule out such a possibility? Single-crystallinity for the entire LIPSS is a strong claim that needs more evidence.

(iv) It seems that Figure 5 caption and the text below it describe the solid and dashed lines differently.

(v) According to Ref. 41, c_m should be $a_r \cdot c_r$. The figure in Ref. 47 gives $c_m \approx -a_r \cdot c_r$. If the tetragonal symmetry of the rutile phase is used to reassign a_r and b_r , then c_m should be $\pm b_r \cdot c_r$. Strictly speaking, the axis relation on page 8 should use an additional minus sign.

(vi) Do the different crystal orientations indicated in Figure 6a mean the presence of twin structures within one crystal? Do the authors need to adjust their description of single crystals accordingly?

(vii) Another major issue of the manuscript is the proposed model of LIPSS formation on page 9. The authors give no references for this well-studied material to support their hypothesis. Are the authors aware of any VO_2 transient reflectivity (TR) or absorption (TA) reports that described such polarization-dependent anisotropy observed within 1 ps following photoexcitation? The review of Ref. 47 did not mention such finding. *Nat. Commun.* 6:6849 (2015) shows results for probe polarization dependence but at low fluences. In it, essentially no difference was observed within the first 1 ps. If there were anisotropy in the induced dielectric function, transient optical studies should have picked up the finding. *Nat. Commun.* 11:5770 (2020) did not mention anisotropy of transient dielectric constant either, although possibly due to the fundamental field-polarization principle of the PINEM effect.

If the authors are not able to find some reference support or show supporting TR or TA results using their instrument, the speculation on page 9 would very likely be incorrect.

(viii) The authors strongly emphasize single-crystalline surface structures in the beginning of page 10 and the following. However, polycrystalline or preferentially oriented VO_2 is still switchable as long as the stoichiometry of VO_2 is kept. A sharp hysteresis is still possible as seen often in the literature. I don't see the reason for such emphasis.

Reviewer #2

(Remarks to the Author)

In the view of the fact that laser-induced materials often become amorphous or polycrystalline, and are rarely re-writable or switchable, this manuscript experimentally proposes self-organized laser-induced surface periodic structures by utilizing the characteristic of metal-to-insulator transition of vanadium dioxide. In addition, the cause of the subharmonic structure is further investigated. The manuscript appears to be well organized.

In general, I would support the publication of this work, provided that the authors could respond to the following comments and questions.

1. For the appearance of the high-spatial-frequency ripples in lower fluences (Fig. 4b), the authors explained that it is the combination of lower photon-doping and a lower amount of metallic VO_2 that opens up another self-stabilizing channel for surface plasmon polariton formation. Could you explain it in a little more detail?
2. Comparing with conventional groove pattern, the subharmonic structure is independent of the polarization, under which the single-crystal surface can have different oriented domains, as shown in Fig. 6. The subharmonic-structure effect is much interesting. However, the reason why the writing fluence of 90 mJ/cm², not 120 mJ/cm² or more, yields the subharmonic structure is not clear. Therefore, I recommend to explain it.
3. For Fig. 6(a), it only includes the results of the subharmonic structure, in comparison to the mixture of conventional and subharmonic ones (Fig. 5). On page 8, the authors stated that they searched the area with two domains and wrote laser grooves. I am wondering whether the conventional and subharmonic structures can be completely separated.
4. The temporary state in which $\epsilon_{\text{par}} < 0$ and $\epsilon_{\text{perp}} > 0$ is critical to the directionality of the sub-harmonic grooves. However, how about the lasting time for the transient regime. In other words, the switching time of the metal-to-insulator transition is 80 fs, shorter than the pulse duration of 300 fs. Hence, I feel confused what causes the transient regime, and why so short transition time results in the groove.

Reviewer #3

(Remarks to the Author)

The authors present a research paper with the first experimental report on the generation of periodic surface structures induced by femtosecond laser pulses on single crystal samples of Vanadium Dioxide. The laser system used operates at 1030 nm and delivers pulses of 300 fs duration (nominal values) that are ultimately responsible for the formation of the

periodic structures reported. The novelty here lays in the generation of periodic structures that follow one crystalline axis preferentially, regardless of the polarization orientation, constituting one of the first experimental demonstrations in the field that demonstrates the importance and relevance of crystal structures and orientations. I consider the paper can be considered promptly for publication in Nature Communications after some minor corrections.

- I would like to ask the authors if the presented experiments were also studied with a polycrystalline sample of the same material. In that way, a confirmation on the crystal orientation dependence can be demonstrated without doubt. I would expect the larger period not to form since it needs the crystalline structure to be developed. In case this has been tested, please, add a comment on that line.

- The roughness of the surfaces used for the experiments is listed as below 10 nm, but there is no indication of how this value was achieved or measured. It is known that scattering points strongly influences the direction of the formed periodic structures since it directs the formed interference between the plasmon at the surface and the remaining part of the pulse. Please, comment on that in the paper.

- In Fig. 4, it should be indicated the region where the structures in D-F were acquired. Since the spatial profile used for the experiments was Gaussian, the local fluence is different in the showed areas. It can be noticed for example, that Fig. 4 e contains two different structures, on the right the period is larger than on the structures on the left. Was that one inset acquired close to the edge of the spot in Fig. 4B?

Version 1:

Reviewer comments:

Reviewer #1

(Remarks to the Author)

The authors have addressed all of my comments, made the corresponding changes, and added informative information. I recommend the manuscript to be accepted for publication.

As a minor point, please comment on the horizontal dotted line in Figure S2. Is it intended?

Reviewer #2

(Remarks to the Author)

The authors have made careful revisions to this nice work. I recommend the publication of this work as it is.

Reviewer #3

(Remarks to the Author)

In this new version of the paper, the authors accurately address the comments from the reviewers, paying particular attention to the characterization techniques implemented, and the details regarding the possible errors in the measurements of the structures orientation. Additional data is included in the new version which provides solid background for the conclusions presented. In the current state, I could propose the paper to be accepted as it is in Nature Communications.

Reviewer 1

The manuscript by Kazenwadel et al. describes intriguing observations of laser-induced periodic surface structures (LIPSS) on VO₂ single crystals. The authors examine the build-up of the LIPSS after a number of laser shots, the orientation of the generated grooves relative to the laser polarization, the connection between their spatial periodicity and the laser fluence used, and the groove structure's capability to change the polarization of incoming light. The authors further report the rewritability of the LIPSS in different directions and find a different subharmonic structure registered to a specific crystal axis. The authors claim that the LIPSS remain single-crystalline based on electron back-scatter diffraction (EBSD) and hypothesize the presence of an anisotropy in the dielectric function of the transient metallic phase that could facilitate surface plasmon polariton formation for the LIPSS. With the produced grating-like structure, the authors suggest potential applications such as active metamaterials on macroscopic scales.

In my opinion, this is an important discovery with both fundamental and technological significance. The report appeals to broad readership from materials and engineering research fields, including those highly interested in the use of the correlated material VO₂. The results are reasonably analyzed and explained with a proposed mechanism for the LIPSS formation. Overall, I believe it is suitable for consideration of publication. However, there are a few major issues that need to be resolved first.

Thank you for this positive review.

The authors mention the highest intensity observed at ~45° on page 3 of the text but Figure 1d shows only the observations made with a 30° step size not covering the claimed angle.

We changed the discussion from highest to lowest intensity (90° and 0°) which is a value that is precisely covered in the experiment. We also adapted the text to better explain the data in Fig. 1d.

The authors' language of "periodic surface structures aligned orthogonal to the polarization of the laser light" on page 2 appears to be very specific. However, upon a closer look at Figure 2, the grooves' tile angles vary noticeably from the left side to the right. While the preferred orientation is clear, the results are not strictly like the authors' claim. Figure 1c also show a certain angular range in each panel. Therefore, I suggest the authors to adjust their description.

Indeed, the grooves are not aligned perfectly, and we report the error now in the figure caption and text. The width is $\pm 15^\circ$, calculated from the Fourier transforms shown in Fig. 5.

The EBSD patterns in Figures 6 and S1 are used to claim the single-crystallinity of the LIPSS. However, there are Kikuchi bands that are not indicated in the figures; are they also produced from the same crystal orientation or actually by maybe twin structures? Do the authors have more evidence for single crystallinity? Would it be just preferred orientations? Figure 1b appears to show features of grains/islands/bumps that may be hard to consider as fully single-crystalline, unless they were produced as a result of materials indeed being ablated (the authors describe this probability on top of page 3) and no melting and then recrystallization (the authors mention partial melting on page 4).

Here I provide another possibility that the grooves are produced with some contributions from materials fast melting and recrystallization, producing a certain thickness of polycrystalline surface layer. Given that EBSD information depth is tens of nanometers, this alternative scenario can still produce the observed pattern, with the polycrystalline layer giving diffuse scattering onto the ordered pattern from the deeper part of the groove structure. How would the authors rule out such a possibility? Single-crystallinity for the entire LIPSS is a strong claim that needs more evidence.

We have now re-measured the EBSD data with the pattern orthogonal to the plane of the incidence of the electron beam. The electron penetration depths at 20 keV is ~20 nm, the height of the ridges is ~400 nm, and the incidence angle of EBSD is 70°. We therefore now probe the ridges alone. The updated Fig. S1 shows clearly a single-crystal result, confirming that the ridges are mostly single-crystalline. All visible features can be indexed properly but not all labels are plotted in the figure for clarity. The manuscript is updated correspondingly.

It seems that Figure 5 caption and the text below it describe the solid and dashed lines differently.

Thank you; the caption and text are now updated to match.

According to Ref. 41, c_m should be $a_r - c_r$. The figure in Ref. 47 gives $c_m \approx -a_r - c_r$. If the tetragonal symmetry of the rutile phase is used to reassign a_r and b_r , then c_m should be $\pm b_r - c_r$. Strictly speaking, the axis relation on page 8 should use an additional minus sign.

We agree, the minus sign (initially removed for simplicity) is now included everywhere in the text.

Do the different crystal orientations indicated in Figure 6a mean the presence of twin structures within one crystal? Do the authors need to adjust their description of single crystals accordingly?

Yes, we agree. All our crystals are single-crystalline in the high temperature phase but, depending on the cooling procedure, some of them indeed form differently oriented twin domains in the low-temperature phase (see for example Fig. 6a). We clarify this now in the text.

Another major issue of the manuscript is the proposed model of LIPSS formation on page 9. The authors give no references for this well-studied material to support their hypothesis. Are the authors aware of any VO₂ transient reflectivity (TR) or absorption (TA) reports that described such polarization-dependent anisotropy observed within 1 ps following photoexcitation? The review of Ref. 47 did not mention such finding. /Nat. Commun./ 6:6849 (2015) shows results for probe polarization dependence but at low fluences. In it, essentially no difference was observed within the first 1 ps. If there were anisotropy in the induced dielectric function, transient optical studies should have picked up the finding. /Nat. Commun./ 11:5770 (2020) did not mention anisotropy of transient dielectric constant either, although possibly due to the fundamental field-polarization principle of the PINEM effect.

If the authors are not able to find some reference support or show supporting TR or TA results using their instrument, the speculation on page 9 would very likely be incorrect.

Unfortunately, almost all available transient reflectivity and transient absorption measurements cannot see crystal anisotropy, because they are conducted on thin films or polycrystalline samples^{1,2} [25]. Even some bulk samples [42,55] have domain structures of differently oriented low-temperature phases, and therefore the anisotropy averages out.

Nat. Commun. 6:6849 (2015) [27] is one of the rare papers that used single-crystalline samples and looked at the probe polarisation dependence. The crystals used there have the same orientations as ours due to the similar sample preparation [54]. Unfortunately, the full time-dependent data is only shown for two polarisations, parallel and orthogonal to the high-temperature c_r axis. However, the

¹ A.. Pashkin, C. Kübler, H. Ehrke, R. Lopez, A. Halabica, R.F. Haglund, R. Huber, A. Leitenstorfer, Ultrafast insulator-metal phase transition in VO₂ studied by multiterahertz spectroscopy, Phys. Rev. B 83 (2011) 195120. <https://doi.org/10.1103/physrevb.83.195120>.

² C. Kübler, H. Ehrke, R. Huber, R. Lopez, A. Halabica, R.F. Haglund, A. Leitenstorfer, Coherent structural dynamics and electronic correlations during an ultrafast insulator-to-metal phase transition in VO₂, Phys. Rev. Lett. 99 (2007) 116401. <https://doi.org/10.1103/physrevlett.99.116401>.

optical anisotropy in the low temperature phase is along the low-temperature \mathbf{c}_m axis, which is aligned at an angle of 123° [42] to \mathbf{c}_r . Together with the 3D-rotated unit cell in their sample (compare insets in our Fig. 6a) gives a projection of $\sim 41^\circ$. Therefore, a measurement oriented parallel or orthogonal to \mathbf{c}_r can only catch a small projection of the full anisotropy. The full rotational data at 3 ps is too late for transient anisotropy because electrons and phonons have already thermalized. Also, the crystals in Nat. Commun. 6:6849 (2015) are nanocrystals, smaller than the laser beam, washing out potential anisotropy. We cite Nat. Commun. 6:6849 (2015) [27] now in the main text and explain these connections.

The other mentioned paper, Nat. Commun. 11:5770 (2020), measures PINEM on a nanowire with unclear crystallinity and neither attempts nor performs anisotropy measurements.

In the revised manuscript, we now report an optical anisotropy measurement from our single-crystalline material (see new extended data figure S2). We observe a strong optical anisotropy in exactly the same direction as the sub-harmonic grooves. When we photodope by the laser [25], conductivity increases, but the different initial values imply different final values, at least for some time, before the transition is complete and electrons and phonons thermalize (~ 300 fs). During some of that time, the materials is metallic in one direction and non-metallic in the other one. Therefore, we produce canalized light [57-60]. In the revised main text, we now explain more clearly this line of argument.

The authors strongly emphasize single-crystalline surface structures in the beginning of page 10 and the following. However, polycrystalline or preferentially oriented VO₂ is still switchable as long as the stoichiometry of VO₂ is kept. A sharp hysteresis is still possible as seen often in the literature. I don't see the reason for such emphasis.

There is evidence [61-63] that smaller grain sizes substantially broaden the hysteresis curves. In the revised manuscript, we now cite the corresponding literature and clarify our point.

Reviewer 2

In the view of the fact that laser-induced materials often become amorphous or polycrystalline, and are rarely re-writable or switchable, this manuscript experimentally propose self-organized laser-induced surface periodic structures by utilizing the characteristic of metal-to-insulator transition of vanadium dioxide. In addition, the cause of the subharmonic structure is further investigated. The manuscript appears to be well organized.

In general, I would support the publication of this work, provided that the authors could respond to the following comments and questions.

Thank you for this positive review.

1. For the appearance of the high-spatial-frequency ripples in lower fluences (Fig. 4b), the authors explained that it is the combination of lower photon-doping and a lower amount of metallic VO₂ that opens up another self-stabilizing channel for surface plasmon polariton formation. Could you explain it in a little more detail?

These high-frequency fringes are indeed interesting but not the central topic of our report. They probably do not originate from higher-harmonic generation³ [52] because they appear for rather low laser powers in our experiment. We think that they may originate from some kind of plasmon formation [51] potentially influenced by the phase transformation of VO₂. In the text, we now discuss these possibilities more clearly, but also mention the speculative nature of these thoughts.

2. Comparing with conventional groove pattern, the subharmonic structure is independent of the polarization, under which the single-crystal surface can have different oriented domains, as shown in Fig. 6. The subharmonic-structure effect is much interesting. However, the reason why the writing fluence of 90 mJ/cm², not 120 mJ/cm² or more, yields the subharmonic structure is not clear. Therefore, I recommend to explain it.

We added that at higher powers, the conventional phenomenon dominates the pattern formation. We also explain and discuss now in more detail the three creation mechanisms (compare other reviewer requests).

3. For Fig. 6(a), it only includes the results of the subharmonic structure, in comparison to the mixture of conventional and subharmonic ones (Fig. 5). On page 8, the authors stated that they searched the area with two domains and wrote laser grooves. I am wondering whether the conventional and subharmonic structures can be completely separated.

In the revised manuscript, we now explain that we do not expect that the two phenomena can be completely disentangled.

4. The temporary state in which $\epsilon_{\text{par}} < 0$ and $\epsilon_{\text{perp}} > 0$ is critical to the directionality of the sub-harmonic grooves. However, how about the lasting time for the transient regime. In other words, the switching time of the metal-to-insulator transition is 80 fs, shorter than the pulse duration of 300fs. Hence, I feel confused what causes the transient regime, and why so short transition time results in the groove.

In the revised manuscript, we now report an optical anisotropy measurement from our single-crystalline material (see new extended data figure 2). We see a strong anisotropy of the low temperature phase along the monoclinic c_m axis. When the laser hits the sample, the material is photodoped [25], increasing the conductivity homogeneously. As a result, at least for some time, the initial anisotropy will persist and we get the transient regime of canalized light (compare response to reviewer 1). We reworked the corresponding section in the paper and added a supplementary figure that explains our suggested explanations in more detail.

Reviewer 3

The authors present a research paper with the first experimental report on the generation of periodic surface structures induced by femtosecond laser pulses on single crystal samples of Vanadium Dioxide. The laser system used operates at 1030 nm and delivers pulses of 300 fs duration (nominal values) that are ultimately responsible for the formation of the periodic structures reported. The novelty here lays in the generation of periodic structures that follow one crystalline axis preferentially, regardless of the polarization orientation, constituting one of the first experimental demonstrations in the field that demonstrates the importance and relevance of

³ X. Shi, X. Xu, Laser fluence dependence of ripple formation on fused silica by femtosecond laser irradiation, Appl. Phys. A 125 (2019) 256. <https://doi.org/10.1007/s00339-019-2554-4>.

crystal structures and orientations. I consider the paper can be considered promptly for publication in Nature Communications after some minor corrections.

Thanks a lot for this positive review.

- I would like to ask the authors if the presented experiments were also studied with a polycrystalline sample of the same material. In that way, a confirmation on the crystal orientation dependance can be demonstrated without doubt. I would expect the larger period not to form since it needs the crystalline structure to be developed. In case this has been tested, please, add a comment on that line.

Unfortunately, we have only polycrystalline thin-films available, but those burn at these high fluences. We hope that the clear dependence on low-temperature domain data depicted in Fig. 4a shows the correlation with the crystal axis clear enough.

The roughness of the surfaces used for the experiments is listed as below 10 nm, but there is no indication of how this value was achieved or measured. It is known that scattering points strongly influences the direction of the formed periodic structures since it directs the formed interference between the plasmon at the surface and the remaining part of the pulse. Please, comment on that in the paper.

Scanning electron microscopy data from our crystal surfaces (see Ref. 43 in our revised paper, and below) shows no discernible features except one big crack in the middle, used to optimize resolution and contrast. We therefore deduce a roughness of ~10 nm as an order of magnitude. In the revised manuscript, we now say more clearly how these roughness values are inferred.

[REDACTED]

- In Fig. 4, it should be indicated the region where the structures in D-F were acquired. Since the spatial profile used for the experiments was Gaussian, the local fluence is different in the showed areas. It can be noticed for example, that Fig. 4 e contains two different structures, on the right the period is larger than on the structures on the left. Was that one inset acquired close to the edge of the spot in Fig. 4B?

Your assumptions concerning Fig. 4e are correct. In the revised paper, we now added boxes that define the regions of the zooms.

Editorial

As requested, additional experiments and new data (backscattering diffraction and optical anisotropy) now strengthen our results.